# Auditing Empirical Privacy Protection for Adaptations of Large Language Models

**Lorenzo Rossi, Bartłomiej Marek, Vincent Hanke, Xun Wang,**
**Michael Backes, Adam Dziedzic, Franziska Boenisch**
CISPA Helmholtz Center for Information Security

*Abstract*—Recent work has applied differential privacy (DP) methods to adapt large language models (LLMs) for sensitive applications. While DP offers theoretical privacy guarantees, their practical implications for LLM adaptations remain uncertain. This uncertainty arises from LLM pretraining, where overlap and interdependencies between pretraining and adaptation data can impact privacy leakage despite DP adaptation efforts. To analyze the issue from a practical standpoint, we thoroughly investigate privacy risks under "private" adaptations in LLMs. Relying on the latest privacy attacks, such as robust membership inference, we study the actual privacy risks for the pretraining and adaptation data. We benchmark the privacy risks by systematically varying the distribution of adaptation data, ranging from data perfectly overlapping with the pretraining set through in-distribution (IID) scenarios to entirely out-of-distribution (OOD) examples. Additionally, we evaluate how different kinds of adaptation methods and different privacy regimes impact the vulnerability. Our results reveal that distribution shifts significantly affect the vulnerability to privacy attacks: the closer the distribution of the adaptation data is to the pretraining distribution, the higher its practical privacy risk, even when there is no overlap between pretraining and adaptation data. We find that the highest empirical privacy protection is achieved for OOD data using parameter-efficient fine-tuning (PEFT) methods, such as LoRA. Surprisingly, when considering data from the same distribution, using the pertaining data for adaptations exhibits a similar privacy leakage as the corresponding validation data. To effectively prevent privacy leakage, it is required to train the adaptations with strict differential privacy protection (with $\varepsilon < 0.1$). Finally, our results show that private adaptations, especially done with prefix tuning, can also decrease the empirical leakage from the pretraining data.

## 1. Introduction

The use of large language models (LLMs) for sensitive downstream tasks has grown rapidly, often implemented through methods that guarantee differential privacy (DP) [1, 2] for the adaptation data [3, 4, 5, 6, 7]. However, this approach may not provide the anticipated privacy protections [8]. The challenge arises from potential overlap or complex dependencies between data used to pretrain the LLMs and the adaptation dataset and from the leakage of the pretraining data that is not protected through the

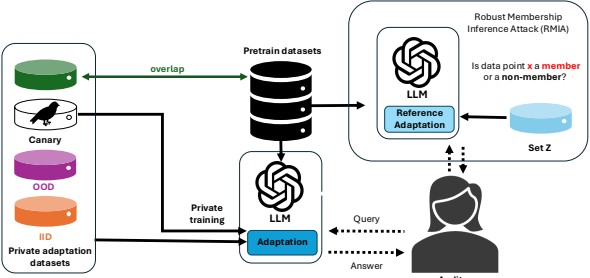

Figure 1: **Setup for Privacy Auditing of LLM Adaptations.** We consider different types of datasets used for LLM adaptations ranging from perfect overlap to OOD data with respect to the pretraining data. The auditor is assumed to have query access to the LLM after its adaptation. To instantiate the membership inference attack, we use one reference model that is fully fine-tuned on $Z$, where $Z$ is a set of samples from the same distribution as the adaptation data.

differentially private adaptations. The problem is exacerbated by the fact that for most large models, their pretraining datasets are not disclosed, rendering a structured study of the interdependencies with the private adaptation data impossible.

**Contributions.** In this paper, we examine practical privacy risks that arise under "private" LLM adaptations. Therefore, we first systematically characterize the setup of privacy auditing required for the novel *pretrain-adapt* learning paradigm that underlies LLMs and their adaptations. We identify four different stages of auditing under the pretrain-adapt paradigm, namely (1) audit pretraining, (2) audit adaptations, (3) joint audit of pretraining and adaptations, and (4) post-adaptation auditing of the pretraining. A general overview of privacy auditing for adapted LLMs is provided in Figure 1. Second, we re-define the *membership inference game* for each of the identified stages to provide a formal grounding necessary for structured privacy audits. We then instantiate multiple *membership inference attacks* [9, 10] against adapted LLMs to audit privacy leakage from the adaptations, and to understand the impact of adaptations on the pretraining privacy, corresponding to stage (2) and (4) in our taxonomy on privacy audits in the pretrain-adapt paradigm.

We systematically analyze a spectrum of possible distributions for the adaptation data with respect to the pretraining

data—from perfect OOD examples—to understand the possible privacy implications over all setups. We also study a wide range of private adaptation methods and different privacy regimes for structured reasoning about the resulting risks. Finally, we discuss the way and the challenges towards a wholistic privacy auditing of adapted LLMs in the pretrain-adapt paradigm.

**Summary of our Empirical Findings.** Our results empirically confirm the theoretical concern that pretraining significantly impacts the privacy risk of the adaptation data [8]. Especially the closeness of pretraining and adaptation data distributions plays a crucial role: the closer the adaptation data distribution is to the pretraining data, even when there is *no overlap* in the datasets, the higher the privacy risks. Also, the type of (private) adaptation has a significant impact on privacy leakage. We find that parameter-efficient fine-tuning (PEFT) methods, such as LoRA [11] with their DP version [4], even though they yield utility on par with fine-tuning approaches, cause significantly lower empirical privacy leakage. Thus, the best empirical privacy protection can be achieved for adaptations with PEFT methods on OOD data. Interestingly, when adapting on data from the same distribution, privacy leakage levels are similar between adaptation data and the corresponding validation data. To effectively prevent privacy leakage, adaptations must be trained with stringent differential privacy constraints ($\varepsilon < 0.1$). Finally, we find that (privately) adapting pretrained models with prefix tuning [12, 13] using their DP versions [6, 14] can also reduce the leakage of the pretraining data. This is probably because the noise added for the private adaptation adds protection to the pretraining data [15].

## 2. Background

We begin with a background on DP, DP adaptations for LLMs, and MIA.

### 2.1. Differential Privacy

The mathematical framework of DP [1] formalizes the intuition that privacy guarantees can be obtained when a randomized mechanism $\mathcal{M}$ executed on two neighboring datasets $D$, $D'$ that differ in only one data point, yields roughly the same result, *i.e.,*

$$\Pr[\mathcal{M}(D) \in S] \leq e^\epsilon \cdot \Pr[\mathcal{M}(D') \in S] + \delta. \quad (1)$$

The privacy parameter $\varepsilon$ specifies by how much the result is allowed to differ and $\delta$ is the probability of failure to meet that guarantee. There are two canonical algorithms to implement DP guarantees in machine learning (ML): DPSGD (the *Differentially Private Stochastic Gradient Descent*) algorithm [16], which extends standard stochastic gradient descent with clipping and noising gradients, and PATE (*Private Aggregation of Teacher Ensembles*) [17, 18], which is an inference time algorithm that privately transfers knowledge from an ensemble of teachers to a public student model.

## 2.2. Private Adaptations of LLMs

LLMs are pretrained on extensive amounts of public data, followed by their adaptation to private downstream tasks. The existing methods for private LLM adaptations fall into two categories: (1) *private tuning methods*, such as PrivateLoRA [4] or PromptDPSGD [6], that rely on access to the LLM gradients and are based on the DPSGD algorithm, and (2) *private in-context learning (ICL) methods*, such as DP-ICL [19] or PromptPATE [6], which require only API (black-box) access to the LLM and are based on the PATE algorithm. The private tuning-based methods can be applied only to LLMs that expose their parameters, which are commonly referred to as *open LLMs*. On the contrary, the private ICL techniques are applicable to both open and closed LLMs, such as GPT4 [20] or Claude3 [21]. However, individual users have to share their private data with the LLM providers to perform adaptations of the closed LLMs. Given this additional privacy leakage and the fact that it was recently shown that adaptations on open LLMs strictly outperform their closed counterparts in terms of privacy protection, performance, and price [14], in our work, we rely on the private adaptations of *open LLMs*.

There are three main approaches to private adaptations of open LLMs: prompt-based, parameter-efficient fine-tuning, and full fine-tuning.

**(1) Prompt-based adaptations** introduce a small number of additional parameters, usually comprising less than 1% of the total LLM parameters, which are applied only in the input space of the model. These parameters may be added at the level of token embeddings (soft prompts [22, 23]) or to all (attention) layers of the LLM (prefix-tuning [12, 24]). [6] proposed *PromptDPSGD*, which adapts the differential private stochastic gradient descent (DPSGD) algorithm [16] for use with soft prompts. A key advantage of prompt-based adaptations is their ability to support multi-task batch processing, meaning multiple soft prompts for different tasks (and users) can be handled in the same mini-batch during training or inference.

**(2) Parameter-efficient fine-tuning-based adaptations** introduce a slightly larger number of parameters, typically under 10% of the total LLM parameters, which are placed within the model, often in each block of a transformer [25]. These added parameters are tuned while keeping the original pretrained parameters frozen. *PrivateLoRA* [4] extends LoRA [11] with DP guarantees by leveraging the DPSGD algorithm.

**(3) Full fine-tuning** involves fine-tuning either the entire model or the last few layers (the latter is also referred to as "head fine-tuning"). *DP-FineTune* [5], which also relies on the DPSGD algorithm, demonstrates that full fine-tuning with DP optimization can offer strong privacy guarantees while maintaining good performance. The general trend in selecting an appropriate method suggests that more complex tasks require a higher number of tunable parameters [6]. For simple tasks, PromptDPSGD [6] is often sufficient, while DP-LoRA [4] is recommended for tasks of moderate difficulty, and full fine-tuning [5] is best suited for complex tasks.

## 2.3. Membership Inference Attacks

A membership inference attack (MIA) [9] aims to determine whether a specific data point can be identified as part of a model's training set. This approach plays a crucial role in applications ranging from privacy assurance [26] to identifying protected or copyrighted content embedded in pretraining data [27]. While most MIA research has focused on supervised learning settings [28], new advancements reveal their broader relevance. [29] revealed a discrete-prompt-based MIA, disclosing vulnerabilities in proprietary LLMs like GPT-3, which risk leaking private information through prompt-based queries [6]. Robust membership inference attack (RMIA) [10] were recently introduced and outperformed prior attacks by optimizing computation with a more precise null hypothesis and leveraging both a reference model and population data. This not only enhances the attack's strength and robustness but also makes the attack computationally more tractable, as it requires only one reference (*shadow*) model where prior work [28] required training hundreds. Due to its strong performance, we mainly harness RMIA in our work to quantify privacy leakage from private adaptations of open LLMs. Min-K% [30] offers a computationally efficient and reference-free method for pretraining data detection in LLMs. By focusing on outlier tokens with low probabilities, this method calculates an average log-likelihood score to determine whether a text was included in the training corpus. We rely on Min-K% for additional privacy evaluations. See Appendix A for a more in-depth description.

## 3. Characterizing Privacy Audits under the Pretrain-Adapt Learning Paradigm

In standard ML, where models are trained from scratch on a given, potentially sensitive, dataset, privacy audits can be conducted directly with respect to the training data. However, under the pretrain-adapt learning paradigm, we find that privacy audits are significantly more complex due to the availability of both the pretraining and the adaptation dataset and their interplay. In this section, we taxonomize the different stages relevant for auditing under the pretrain-adapt paradigm and define auditing of the different stages as adversarial games that allow us for a structured reasoning about leakage and privacy protection.

### 3.1. Taxonomizing Audit Stages

We identify four stages of auditing based on the learning paradigm and its respective pretraining dataset $S$ and adaptation data $D$ in our taxonomy.

**(1) Auditing pretraining** is the most similar stage to the standard ML auditing. It aims at identifying privacy leakage of the pretraining data from the pretrained model. The difference to privacy audits in standard ML is that the pretraining is usually done on much larger datasets, with larger models where the applicability and effectiveness of rigorous privacy protection through DP [31], as well as the applicability of standard privacy auditing techniques like MIA [32], are limited.

**(2) Auditing adaptations** is a new aspect in the pretrain-adapt paradigm. It is concerned with detecting leakage of the adaptation dataset from the adapted LLM. The key differentiating factor to privacy audits in standard ML is using a pretrained model that the adaptations are trained on instead of a random initialization. We assume the same pretrained model is used for all the considered adaptations in an adaptation audit.

**(3) Joint Auditing of Pretraining and Adaptations** considers both stages of pretraining and adaptations together. The goal is to audit leakage from both the pretraining and adaptation set from the adapted LLM. Under privacy preservation, the most standard setup consists of a non-DP-trained LLM and DP-trained adaptations.

**(4) Auditing Pretraining Post-Adaptations** evaluates how the (private) adaptations influence the potential protection of the data points used for pretraining, which is usually conducted without any formal guarantees. Changes to the model behavior induced through adaptations or noise added during their training might influence the effective exposure of pretraining data from model predictions.

### 3.2. Defining Audits as Adversarial Games

Privacy audits can be modeled as an *adversarial game* $\mathcal{G}$ [33, 34] where the main task is to guess if a given data point $x$ was in a model's training set or not. This game can, therefore, also be referred to as the *membership inference game*. For standard ML, it is formulated as follows:

**Standard ML Adversarial Game.** Given a dataset $D$, the target sample $x$, a training algorithm $\theta \xleftarrow{\text{T}} D$, a sampling procedure $x \xleftarrow{\text{R}} \{0,1\}$, $\mathcal{G}$ is executed as:

1) The challenger samples a binary variable $a \xleftarrow{\text{R}} \{0,1\}$ uniformly at random
2) The challenger trains a model $\theta \xleftarrow{\text{T}} \tilde{D}$, where $\tilde{D} = D$ if $a = 0$, otherwise $\tilde{D} = D \cup \{x\}$
3) The challenger sends $\theta$ to the attacker
4) The attacker guesses $\hat{a} \leftarrow \mathcal{A}(\theta, x)$

The attacker wins if his guess $\hat{a}$ on whether $x$ was used to train $\theta$ is correct.

To audit the different stages of the pretrain-adapt paradigm, we need to define a new game that accounts for the existence of both the pretraining dataset $S$ and the adaptation data $D$.

**Pretrain-Adapt Adversarial Game.** We define the adversarial game $\mathcal{G}$ analogous to the one for standard ML, yet take two datasets, $S$ the pretraining data, and $D$ the adaptation data into account. Additionally, we denote the pretraining procedure by $T$ and an adaptation procedure by $T'$. We mark the deviations to the original game in blue.

1) The challenger samples $a \xleftarrow{\text{R}} \{0,1\}$ and $b \xleftarrow{\text{R}} \{0,1\}$ (where $a$ and $b$ are binary variables)

2) The challenger trains a model $\theta \xleftarrow{\text{T}} \tilde{S}, \theta_0$, where $\tilde{S} = S$ if $a = 0$, otherwise $\tilde{S} = S \cup \{x\}$
3) The challenger adapts $\theta$ such that $\theta' \xleftarrow{\text{T}'} \tilde{D}$, where $\tilde{D} = D$ if $b = 0$, otherwise $\tilde{D} = D \cup \{x\}$
4) The challenger sends $\theta'$ to the attacker
5) The attacker guesses $\hat{a}, \hat{b} \leftarrow \mathcal{A}(\theta, \theta', x)$

Whether the attacker has to guess both $\hat{a}, \hat{b}$ and what background knowledge they have, *i.e.,* whether they get access to both $\theta$ and $\theta'$ depends on the auditing stage. We detail the background knowledge and guesses by the attacker—formulated as hypotheses with a null hypothesis $H_0$ and an alternative hypothesis $H_A$—for the four auditing stages from our taxonomy.

**(1) Auditing Pretraining.** In this setting, the challenger releases the pretrained model $\theta$ to the attacker. The attacker's goal is correctly guessing whether $x$ was in the pretraining data $S$. Their guesses $\hat{a}$, are over the random variable $a$.

$$H_0 : a = 0 \qquad H_A : a = 1$$

**(2) Auditing Adaptation.** In this setting, the challenger releases only the adapted model $\theta'$ to the attacker. The attacker does not know whether $x \in S$ or not and considers only the adaptation. Their guesses $\hat{b}$, are, hence, over the random variable $b$.

$$H_0 : b = 0 \qquad H_A : b = 1$$

**(3) Joint Auditing.** In this setting, the challenger releases both the pretrained model $\theta$ and the adapted $\theta'$ to the attacker. Depending on the attacker's background knowledge, we consider three possible cases.

1) The attacker knows that $x \notin S$ and guesses $b$.

$$H_0 : (a, b) = (0, 0) \qquad H_A : (a, b) = (0, 1)$$

2) The attacker knows that $x \in S$ and guesses $b$.

$$H_0 : (a, b) = (1, 0) \qquad H_A : (a, b) = (1, 1)$$

3) The attacker knows that the target sample $x$ is either in both (pretraining and adaptation sets) or neither of them and guesses $(a, b)$.

$$H_0 : (a, b) = (0, 0) \qquad H_A : (a, b) = (1, 1)$$

**(4) Post-Adaptation Auditing.** In this setting, the challenger releases both the pretrained $\theta$ and the adapted $\theta'$. It is known that the target sample $x$ is not in $D$ and the attacker takes a guess on $a$.

$$H_0 : (a, b) = (0, 0) \qquad H_A : (a, b) = (1, 0)$$

In essence, auditing pretraining considers only the pretraining itself. Similarly, auditing the adaptations considers the adaptations themselves. On the other hand, the joint adaptation reasons about both pretraining and adaptation sets. Finally, the post-adaptation auditing is also only for the pretraining set, but the applied adaptation influences the auditing. In this work, we focus mainly on the auditing of adaptations as defined in (2) but, on the way, also provide empirical insights into the post-adaptation leakage (4). Finally, we provide a discussion on ways and challenges towards holistic privacy audits under the pretrain-adapt paradigm that take into account all four stages.

## 4. Assessing Empirical Privacy Risks of Private LLM Adaptations

In the following experiments, we evaluate the empirical privacy risks of LLM adaptations in a "black-box" scenario. We focus on *open LLMs*, *i.e.,* LLMs whose weights are publicly available, since relying on closed LLMs, such as GPT or Claude, for adaptations usually requires sharing the private data with the LLM provider, causing additional privacy risks [14].

Our results reveal the vulnerability of LLM adaptations to privacy leakage, which is higher when the adaptation dataset comes from the same distribution as pretraining datasets. The leakage is slightly lower when there is no overlap in the distribution of the adaptation data with the pretraining data. Furthermore, by experimenting with different LLM adaptation techniques, we demonstrate that this choice also impacts privacy vulnerability across most datasets, with prefix tuning leaking the most for the in-distribution adaptation data and full fine-tuning leaking more than other methods for the out-of-distribution adaptation.

### 4.1. Experimental Setup

**Models.** We focus on the open-source Pythia suite of models [35] and its publicly available pretraining data, the Pile dataset [36], which is an 800GB collection of diverse English-language datasets, including text from sources, such as books, academic papers, or source code repositories.

**Datasets.** We categorize the datasets used in our experiments into **in-distribution (IID)** and **out-of-distribution (OOD)**, depending on their relationship to the pretraining data. IID datasets come from the same distribution as the pretraining data, and we identify two cases: one with a full overlap between pretraining and adaptation data, where we use data directly from the pretraining set for the adaptations, and one with no overlap, where the data is sourced from the corresponding validation set from the pretraining distribution. For the IID datasets, we focus on the following Pile subsets: BookCorpus2, consisting of publicly available books, GitHub, a set of open-source code repositories, and Enron Emails [37], a variety of different emails. In contrast, OOD datasets are derived from a different distribution and do not overlap with pretraining data. The OOD datasets we chose for our experiments are: SAMSum [38], an English-language dialogue summarization dataset, and GermanWiki [39], a large set of German Wikipedia entries. These OOD datasets were selected because of their different degrees of variation from the original distribution of the Pile dataset. Although SAMSum shares the same language (English), its general dialogue format, followed by the dialogue summary, is not

present in the pretraining set. GermanWiki, on the other hand, presents wide syntactic and lexical variation from the pretraining dataset.

**Memorized samples.** Another privacy concern showed in prior work [40] is the memorization of samples during pretraining of an LLM. We analyze how adaptations can reduce the effect of the memorization of pretraining data. The definition of a memorized sample follows $k$-extractability from [40]. Here, we have a prompt $p$ of length $k$ and a suffix $s$. If the generation of a model given prompt $p$ generates exactly $s$, the sequence consisting of $p$ and $s$ concatenated is memorized. Furthermore, we also rely on samples from the Pile reported as memorized in Pythia 2.8B by prior work [41]. This set of memorized samples consists of 505 sequences, and we refer to it as Mem Pile.

**Adaptations.** We evaluate different types of adaptations, including fine-tuning of all model parameters [5], or the last layer (*i.e.,* the head) and PEFT methods, such as LoRA [4, 11] and prefix tuning [6, 22]. Considering a Pythia 1B model, we train 1B parameter for full fine-tuning, 1M for LoRA, 130M for prefix tuning, and 100M for last-layer (head) fine-tuning.

**Membership Inference.** For membership inference, we rely on the latest membership inference attack, RMIA (Robust Membership Inference Attack) [10]. We use its offline version because it is computationally effective and it does not require to train customized reference models for each targeted sample (as in the online version of the attack). We also leverage a single reference model for our experiments, as the authors show strong MIA performance even with only one reference model. We consider different types of reference models. Unless explicitly stated, we focus on using a "shadow" model (adaptation) trained in the same way as the target model, but on a different split of the same fine-tuning data. However, we also consider other models: Pythia-14M, Pythia-160M, Pythia-1B (which we report in the results below), Pythia-2.8B [35] , GPT-neox [42], and GPT-2. RMIA has two hyperparameters, a threshold $\gamma$, and a scaling factor $\alpha$ (see Algorithm 1). The results represent the highest AUC achieved through a grid search to optimize these parameters. For an ablation on the RMIA hyperparameters choice, see Figure 6 in Appendix C. Additionally, we consider another commonly used reference-based MIA called Reference [43], which calibrates the loss of the target model on the target sample by dividing it by the loss of a reference model on the target sample. Finally, we also compare Min-K%, as a reference-less baseline. As with RMIA, we report the highest AUC achieved through a grid search on $K$.

## 4.2. Evaluating Practical Privacy Leakage through Membership Inference

We begin by assessing privacy leakage across various LLM adaptation methods. We present the control setup in Figure 2, where all adaptations are trained with $\varepsilon = 8$, share the same learning rate (LR = 0.0001) and the number of epochs (20). Privacy leakage is measured by the AUC score

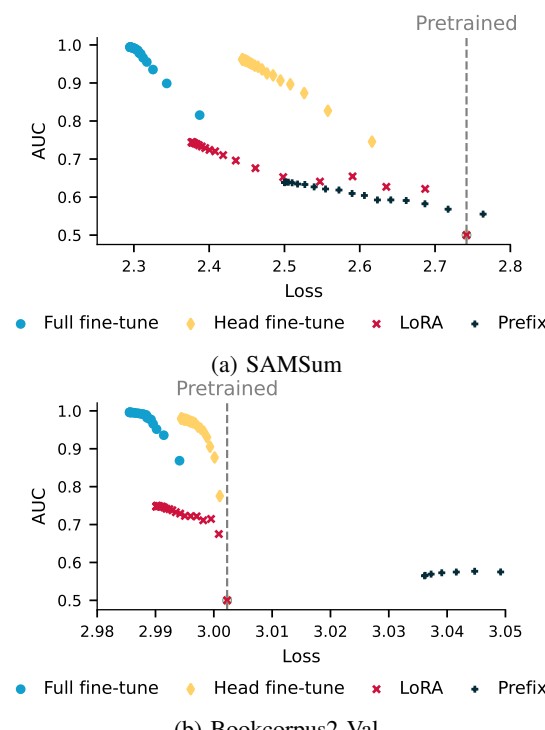

(a) SAMSum

(b) Bookcorpus2 Val

Figure 2: **LoRA exhibits the lowest privacy leakage for a given loss value.** The other methods: Full fine-tune and Head find-tune leak much more. Prefix Tuning leaks the least for OOD (SAMSum) data, however, it is an exception for IID (Bookcorpus2 Val). The x-axis shows the evaluation loss. The y-axis represents the AUC score. All adaptations have been trained with $\varepsilon = 8$, a learning rate of 0.0001, and over 20 epochs.

and plotted against validation loss, representing different stages of model training. Each point in the graph reflects the metric values at the end of a training epoch, with the rightmost lower points marking the start of training. As training progresses, points shift upward and to the left.

The results indicate that for the OOD adaptation dataset (SAMSum), privacy leakage is the highest with Head fine-tuning, followed by Full fine-tuning, for any given loss value. In contrast, LoRA and Prefix Tuning demonstrate significantly lower privacy leakage, with Prefix Tuning yielding the lowest AUC scores. We observe a similar trend with the IID BookCorpus2 dataset: privacy leakage is again the highest for Head fine-tuning, followed by Full fine-tuning, with LoRA consistently showing the lowest AUC scores across the board. An exception occurs with Prefix Tuning on this dataset, where it is more sensitive to parameter settings and incurs a much higher loss than other methods, making direct comparisons challenging. To address this issue, next, we shift to a detailed comparison of each method using its optimal parameter settings and present the individual results accordingly.

We compare privacy leakage across the two dataset types, IID and OOD, in Table 1 and Table 2, respectively. Since

TABLE 1: **Membership Inference for OOD Adaptations.** We audit only the adaptations and assume the same pretrained LLM is used for all adaptations. We present the AUC scores obtained with RMIA, reference, and Min-K% MIAs for the Pythia 1B model adapted on different datasets with $\varepsilon \in \{0.1, 8, \infty\}$.

| MIA | Dataset / Adaptation | SAMSum | | | GermanWiki | | | Average | | |
|---|---|---|---|---|---|---|---|---|---|---|
| | | $\varepsilon=\infty$ | $\varepsilon=8$ | $\varepsilon=0.1$ | $\varepsilon=\infty$ | $\varepsilon=8$ | $\varepsilon=0.1$ | $\varepsilon=\infty$ | $\varepsilon=8$ | $\varepsilon=0.1$ |
| RMIA (shadow) | Prefix Tuning | 1.00 | 0.62 | 0.63 | 1.00 | 0.64 | 0.61 | 1.00 | 0.63 | 0.62 |
| | LoRA | 0.86 | 0.69 | 0.50 | 1.00 | 0.59 | 0.66 | 0.93 | 0.64 | 0.58 |
| | Full fine-tune | 1.00 | 0.82 | 0.62 | 1.00 | 0.71 | 0.55 | 1.00 | 0.77 | 0.59 |
| | Head fine-tune | 1.00 | 0.98 | 0.62 | 1.00 | 0.76 | 0.70 | 1.00 | 0.87 | 0.66 |
| | Average | 0.97 | 0.78 | 0.59 | 1.00 | 0.67 | 0.63 | 0.98 | 0.73 | 0.61 |
| RMIA (Pythia 1B) | Prefix Tuning | 0.94 | 0.51 | 0.51 | 0.91 | 0.50 | 0.50 | 0.92 | 0.50 | 0.51 |
| | LoRA | 0.51 | 0.51 | 0.51 | 0.81 | 0.51 | 0.51 | 0.66 | 0.51 | 0.51 |
| | Full fine-tune | 0.94 | 0.51 | 0.51 | 0.98 | 0.51 | 0.51 | 0.96 | 0.51 | 0.51 |
| | Head fine-tune | 0.96 | 0.52 | 0.51 | 0.97 | 0.51 | 0.50 | 0.97 | 0.52 | 0.50 |
| | Average | 0.84 | 0.51 | 0.51 | 0.92 | 0.51 | 0.50 | 0.88 | 0.51 | 0.51 |
| Reference (Pythia 1B) | Prefix Tuning | 0.93 | 0.50 | 0.51 | 0.92 | 0.50 | 0.50 | 0.92 | 0.50 | 0.50 |
| | LoRA | 0.51 | 0.51 | 0.51 | 0.82 | 0.51 | 0.51 | 0.66 | 0.51 | 0.51 |
| | Full fine-tune | 0.94 | 0.51 | 0.51 | 0.99 | 0.51 | 0.50 | 0.96 | 0.51 | 0.51 |
| | Head fine-tune | 0.97 | 0.52 | 0.51 | 0.98 | 0.51 | 0.50 | 0.97 | 0.51 | 0.50 |
| | Average | 0.84 | 0.51 | 0.51 | 0.93 | 0.51 | 0.50 | 0.88 | 0.51 | 0.51 |
| Min-K% | Prefix Tuning | 0.84 | 0.51 | 0.51 | 0.71 | 0.50 | 0.50 | 0.78 | 0.50 | 0.50 |
| | LoRA | 0.51 | 0.51 | 0.50 | 0.61 | 0.51 | 0.51 | 0.56 | 0.51 | 0.51 |
| | Full fine-tune | 0.83 | 0.51 | 0.50 | 0.88 | 0.51 | 0.50 | 0.86 | 0.51 | 0.50 |
| | Head fine-tune | 0.92 | 0.51 | 0.50 | 0.87 | 0.51 | 0.51 | 0.89 | 0.51 | 0.50 |
| | Average | 0.77 | 0.51 | 0.50 | 0.77 | 0.50 | 0.51 | 0.77 | 0.51 | 0.50 |

TABLE 2: **Membership Inference for in-distribution (IID) Adaptations.** We use the same setup as in Table 1.

| MIA | Dataset / Adaptation | Bookcorpus2 Val | | | Bookcorpus2 Train | | | Github val | | | Enron Val | | | Average | | |
|---|---|---|---|---|---|---|---|---|---|---|---|---|---|---|---|---|
| | | $\varepsilon=\infty$ | $\varepsilon=8$ | $\varepsilon=0.1$ | $\varepsilon=\infty$ | $\varepsilon=8$ | $\varepsilon=0.1$ | $\varepsilon=\infty$ | $\varepsilon=8$ | $\varepsilon=0.1$ | $\varepsilon=\infty$ | $\varepsilon=8$ | $\varepsilon=0.1$ | $\varepsilon=\infty$ | $\varepsilon=8$ | $\varepsilon=0.1$ |
| RMIA (shadow) | Prefix Tuning | 1.00 | 0.89 | 0.56 | 1.00 | 0.90 | 0.55 | 1.00 | 0.93 | 0.63 | 1.00 | 0.88 | 0.58 | 1.00 | 0.90 | 0.58 |
| | LoRA | 1.00 | 0.70 | 0.52 | 1.00 | 0.69 | 0.53 | 1.00 | 0.74 | 0.52 | 1.00 | 0.73 | 0.52 | 1.00 | 0.71 | 0.52 |
| | Full fine-tune | 1.00 | 0.75 | 0.77 | 1.00 | 0.75 | 0.76 | 1.00 | 0.78 | 0.80 | 1.00 | 0.91 | 0.66 | 1.00 | 0.80 | 0.75 |
| | Head fine-tune | 1.00 | 0.72 | 0.73 | 1.00 | 0.72 | 0.72 | 1.00 | 0.80 | 0.74 | 1.00 | 0.57 | 0.65 | 1.00 | 0.70 | 0.71 |
| | Average | 1.00 | 0.77 | 0.65 | 1.00 | 0.76 | 0.64 | 1.00 | 0.81 | 0.67 | 1.00 | 0.77 | 0.60 | 1.00 | 0.78 | 0.64 |
| RMIA (Pythia 1B) | Prefix Tuning | 0.91 | 0.56 | 0.51 | 0.97 | 0.57 | 0.50 | 0.96 | 0.54 | 0.52 | 0.98 | 0.54 | 0.51 | 0.95 | 0.55 | 0.51 |
| | LoRA | 0.87 | 0.52 | 0.52 | 0.96 | 0.51 | 0.51 | 0.91 | 0.51 | 0.50 | 0.98 | 0.56 | 0.51 | 0.93 | 0.52 | 0.51 |
| | Full fine-tune | 0.99 | 0.54 | 0.52 | 1.00 | 0.54 | 0.52 | 0.99 | 0.53 | 0.52 | 0.99 | 0.59 | 0.50 | 1.00 | 0.55 | 0.51 |
| | Head fine-tune | 0.96 | 0.57 | 0.52 | 0.99 | 0.56 | 0.51 | 0.99 | 0.65 | 0.52 | 1.00 | 0.54 | 0.50 | 0.99 | 0.58 | 0.51 |
| | Average | 0.94 | 0.55 | 0.52 | 0.98 | 0.55 | 0.51 | 0.96 | 0.56 | 0.51 | 0.99 | 0.56 | 0.51 | 0.97 | 0.55 | 0.51 |
| Reference (Pythia 1B) | Prefix Tuning | 0.93 | 0.56 | 0.52 | 0.97 | 0.57 | 0.50 | 0.97 | 0.53 | 0.51 | 0.97 | 0.54 | 0.50 | 0.96 | 0.55 | 0.51 |
| | LoRA | 0.89 | 0.52 | 0.52 | 0.97 | 0.51 | 0.51 | 0.92 | 0.51 | 0.50 | 0.97 | 0.55 | 0.51 | 0.94 | 0.52 | 0.51 |
| | Full fine-tune | 1.00 | 0.54 | 0.52 | 1.00 | 0.54 | 0.52 | 0.99 | 0.54 | 0.52 | 0.98 | 0.59 | 0.50 | 0.99 | 0.55 | 0.51 |
| | Head fine-tune | 0.98 | 0.57 | 0.52 | 1.00 | 0.56 | 0.51 | 0.99 | 0.66 | 0.50 | 0.99 | 0.54 | 0.50 | 0.99 | 0.58 | 0.51 |
| | Average | 0.95 | 0.55 | 0.52 | 0.98 | 0.55 | 0.51 | 0.97 | 0.56 | 0.51 | 0.98 | 0.55 | 0.50 | 0.97 | 0.55 | 0.51 |
| Min-K% | Prefix Tuning | 0.78 | 0.51 | 0.50 | 0.70 | 0.51 | 0.50 | 0.65 | 0.52 | 0.52 | 0.66 | 0.51 | 0.52 | 0.70 | 0.51 | 0.51 |
| | LoRA | 0.67 | 0.51 | 0.51 | 0.63 | 0.50 | 0.50 | 0.61 | 0.52 | 0.52 | 0.65 | 0.51 | 0.51 | 0.64 | 0.51 | 0.51 |
| | Full fine-tune | 0.87 | 0.51 | 0.51 | 0.82 | 0.50 | 0.50 | 0.77 | 0.52 | 0.52 | 0.78 | 0.51 | 0.51 | 0.81 | 0.51 | 0.51 |
| | Head fine-tune | 0.75 | 0.51 | 0.51 | 0.72 | 0.50 | 0.51 | 0.64 | 0.52 | 0.52 | 0.70 | 0.51 | 0.51 | 0.70 | 0.51 | 0.51 |
| | Average | 0.77 | 0.51 | 0.51 | 0.72 | 0.50 | 0.50 | 0.67 | 0.52 | 0.52 | 0.70 | 0.51 | 0.51 | 0.71 | 0.51 | 0.51 |

membership inference success is highly dependent on the train-test gap, for a fair comparison of the privacy leakage, we ensure similar evaluation perplexities, in particular, similar validation loss values for specific datasets across adaptation methods, see Table 3. Since the GitHub data subset in the Pile [36] is much bigger than Bookcorpus2 or Enron and has a low perplexity (of only 0.6 compared to 1.0 for Enron) it has a much lower validation loss value, as well. We report the AUC scores for the adaptations in Table 1 and Table 2 and their corresponding validation loss at the end of the adaptation's training (see Table 3 from Appendix D).

We observe different trends in the AUC scores from *RMIA (shadow)* across the data distributions. For $\varepsilon = \infty$, both OOD and IID distributions show similar performance, with an average AUC score close to 1 (almost perfect privacy leakage). However, with $\varepsilon = 8$, the results are more varied. For models adapted with the IID datasets at $\varepsilon = 8$, as shown in Table 2, adapting with prefix tuning results in the highest

privacy leakage, with an average AUC score of 0.9, reaching up to 0.93 for GitHub Val. The second-highest leakage for IID data is observed with Full fine-tuning, averaging an AUC score of 0.8. In contrast, for models adapted with the OOD datasets (see Table 1), Head fine-tuning shows the highest leakage, with an average AUC score of 0.87, while prefix adaptation results in the lowest leakage, with an AUC score of 0.63. Across both dataset types, LoRA consistently provides strong privacy protection, with an average AUC score of 0.64 for OOD data and 0.71 for IID.

We further analyze the differences in privacy leakage between the OOD vs IID datasets used for adaptations. We focus on the $\varepsilon = 8$ that strikes a balance between full leakage with $\varepsilon = \infty$ and very small or no leakage for $\varepsilon = 0.1$. For a fair comparison, we take into account similar validation loss values from Table 3. The SAMSum (OOD) dataset has a similar validation loss as the Enron Val (IID) dataset for $\varepsilon = 8$. We consider the strongest MIA with the RMIA

(*shadow*) and their AUC scores from Table 1 and Table 2. For three out of four adaptation methods (Prefix tuning, LoRA, and Full fine-tune) we observe much higher privacy leakage for Enron (IID) than for SAMSum (OOD). The only exception is the Head fine-tune, for which the adaptation with SAMSum exhibits much higher leakage than with Enron. Overall, for most adaptations, the overlap in the distributions between pretraining sets and the adaptation data incurs higher privacy leakage.

Next, we examine data from the same distribution, specifically Bookcorpus2, where its *Train* version was used for pretraining whereas the corresponding *Val* (validation) set was held out. The observed validation loss values in Table 3 for both the training and validation versions of Bookcorpus2 are very similar. Likewise, the reported AUC scores in Table 2 are also almost the same in both cases. Thus, our findings indicate a comparable privacy leakage when using the pertaining data and the corresponding validation data for the adaptations.

Moreover, we also consider the more realistic settings with *RMIA (Pythia-1B)* and *Reference (Pythia-1B)*, where the attacker does not have access to a shadow model and instead uses the (non-adapted) pretrained model as the reference model. In both cases, AUC scores are similar across all tested models, suggesting that there is no gain in using RMIA when shadow models cannot be trained. Moreover, both cases show significantly reduced performance compared to *RMIA (shadow)*. For $\varepsilon = 0.1$, AUC scores drop to near random guessing for both types of datasets. Increasing the privacy budget to $\varepsilon = 8$ slightly improves the MIA performance on IID datasets, with an average AUC score of 0.58, while the OOD datasets remain at random guessing. At $\varepsilon = \infty$ the two MIAs become more effective. LoRA continues to show high privacy protection compared to the other tested adaptations. On IID datasets, LoRA achieves an average AUC score of 0.93, slightly lower than the scores for Prefix Tuning (0.95), Full fine-tune (1.00), and Head fine-tune (0.99). However, for SAMSum and GermanWiki, we observe a higher difference. Here, we have on average 0.66 for LoRA, while the other adaptations reach scores of more than 0.9.

These results confirm the concerns raised by [8], highlighting the risk of privacy leakage even under the application of DP with $\varepsilon = 8$, usually considered protective in prior literature [4, 5, 6, 44].

Lastly, we compare the development of AUC scores during training on IID and overlap data, as shown in Figure 3. Similar to Figure 2, these results display the AUC score at each epoch during training. To better compare IID and overlap data, we adjust the x-axis to represent the loss difference at each training step, calculated as the initial pretraining loss minus the adapted loss at the current training step. This calibration of the x-axis allows us to compare the two dataset types more precisely. With this setup, we evaluate two subsets of the Pile pretraining set: GitHub and BookCorpus2. First, the figures indicate that further adapting a model on IID data does not significantly improve its performance on that data, with the loss decreasing by only a maximum of 0.015 (GitHub with Full fine-tune).

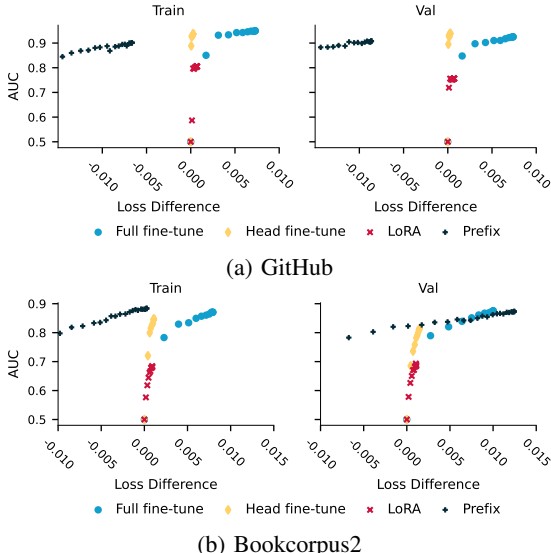

(a) GitHub

(b) Bookcorpus2

Figure 3: **Overlap (Train) and IID data (Val) show the same amount of privacy leakage across training.** The x-axis shows the difference between the initial pretrained loss and the evaluation loss. The y-axis represents the AUC score. All adaptations have been trained with $\varepsilon = 8$.

However, the observed increase in AUC score throughout training shows that the model does learn from the adaptation data.

Additionally, we do not observe any significant difference in privacy leakage or its progression during training between adaptation on the overlap and IID data. A small difference between overlap and IID suggests that dataset inference on the pretraining data, which is currently known to be a complex problem on LLMs pretraining data [45], cannot be easily resolved by analyzing the fine-tuning trajectory loss or the privacy leakage of the fine-tuned data alone.

## 5. Discussion

In the following, we discuss the implications of our findings and the way and challenges towards holistic privacy auditing under the pretrain-adapt paradigm.

### 5.1. Implications of our Findings

Our evaluation highlights a critical trade-off between utility and privacy across different adaptation methods. Despite achieving lower leakage in OOD settings, even the best-performing adaptations like LoRA show vulnerabilities in the scenario of utilizing shadow models with RMIA. Consequently, **privacy adaptations that regard public pretraining data as entirely non-sensitive may unintentionally integrate sensitive information into the models, even when adapted with DP**. This highlights the necessity to perform private LLM adaptations in the high-privacy regime, *i.e.,* with low $\varepsilon$.

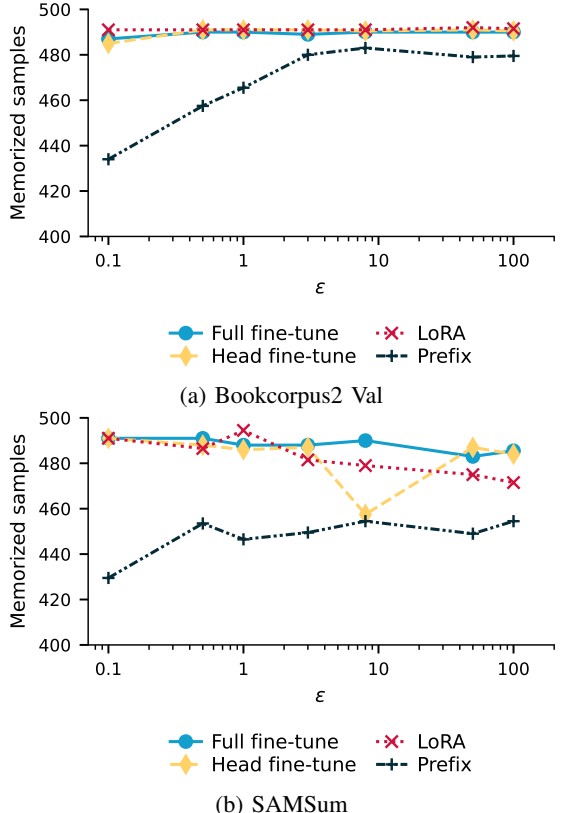

(a) Bookcorpus2 Val

(b) SAMSum

Figure 4: **Fewer memorized samples after prefix tuning.** There are fewer verbatim generations of training samples after the prefix tuning, especially for small $\varepsilon$ values. We present the number of memorized samples from the Pile that remain memorized after adapting Pythia 1B on SAMSum and GermanWiki datasets. The evaluation was done for $\varepsilon = \{0.1, 1, 3, 8, 50, 100, \infty\}$. The x-axis is represented in a log scale.

### 5.2. Towards a Holistic Privacy Auditing for LLMs

In this work, we focus solely on auditing the private adaptations and leakage from pretraining data after adaptations, corresponding to stages (2) and (4) in our taxonomy, respectively. However, for holistic privacy auditing under the pretrain-adapt paradigm, we need ways to audit all stages of the process (jointly).

Given its correspondence to standard ML training—in the sense that it starts from training a randomly initialized model on a given dataset—prior research has attempted to audit pretraining with the same means as used to audit standard ML, namely MIAs, however, with no access to shadow models. Previous work has shown that these new MIAs for LLMs [46, 47] are ineffective for pretraining data [45, 48, 49]. The problem stems from the large pertaining sets (with trillions of tokens) and single training rounds that weaken the membership signal to minimum [50]. The standard MIAs [28] are also not effective in this case. They require training many additional shadow models of similar architecture to

the audited model, which are impractical to train for large models like LLMs due to their size. However, LLMs do memorize some of their training data points [43]. Therefore, [45] proposed an alternative method to membership inference for LLMs based on the established framework of dataset inference [50]. They leveraged the selective combination of many features from many MIAs and aggregated the signals across hundreds or more data points. Future work might leverage these insights to propose better auditing methods for pretraining.

The joint audit of pretraining and adaptation privacy, corresponding to (3) in our taxonomy, represents the largest challenge in practical setups. A main difficulty results from the complex dependencies between pretraining and adaptation data. The problem is exacerbated by the fact that for most LLMs, the pretraining data is either unknown or too large to effectively search through. Thus, advancing effective privacy auditing across all stages of the pretrain-adapt paradigm remains essential to truly privacy-preserving LLMs.

## 6. Conclusions

In this work, we examined the practical privacy risks that arise under "private" adaptations of LLMs within the pretrain-adapt paradigm. We systematically characterized the privacy auditing setup required for this paradigm, identifying four distinct stages: (1) pretraining audits, (2) adaptation audits, (3) joint pretraining and adaptation audits, and (4) post-adaptation audits of pretraining. To enable structured and rigorous privacy audits, we redefined the membership inference game for each stage and instantiated multiple membership inference attacks to assess privacy leakage. Our empirical analysis confirms the theoretical concern that pretraining significantly affects the privacy risks of *adaptation data*. We found that the closeness of adaptation and pretraining data distributions plays a critical role: even in the absence of overlap, higher distributional similarity results in increased privacy leakage. Additionally, we observed that the choice of adaptation method impacts privacy leakage, with PEFT methods, such as LoRA, offering significantly lower privacy risks while maintaining strong utility. When adapting on OOD data, these methods provide the best empirical privacy protection. Furthermore, prefix tuning can reduce the leakage of pretraining data, likely due to the added input noise during private adaptation. Our findings highlight the need for stringent DP constraints (*e.g.*, $\varepsilon < 0.1$) to effectively mitigate privacy risks in LLM adaptations. By providing a comprehensive framework for privacy auditing and uncovering key factors influencing leakage, this work lays a foundation for future research aimed at safeguarding privacy in the pretrain-adapt paradigm.

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

# Appendix

## 1. Membership Inference Attacks

**Min-K%** Min-K% [46] is a recently proposed black-box MIA for large language models. The intuition is that an unseen sample is likely to have low-probability tokens. The MI score is defined as

$$\text{Min-K\%}(x) = \frac{1}{|S|} \sum_{x_i \in S} \log p(x_i | x_1, ..., x_{i-1}), \quad (2)$$

where S is the set of K% tokens with the smallest loss.

**Reference** This approach [43] uses a reference model to calibrate the MI score as follows

$$\text{Ref}(x) = \frac{\mathcal{L}(x|\theta)}{\mathcal{L}(x|\theta_{\text{ref}})}, \quad (3)$$

where $\mathcal{L}(x|\theta)$ indicates the loss of the target sample $x$ on the model $\theta$. $\theta_{\text{ref}}$ represents the reference model used.

**Robust Membership inference attack (RMIA** The adapted RMIA score (Equation (4)) calculation for LLMs for text generation is based on comparing loss values rather than outputs probabilities. For this reason, we have to, instead of comparing prediction probabilities or logits, compare the loss of the target data point against the loss of reference models on population data (Equation (5)) and flip to a minority voting approach, where the decision is based on how much lower the loss of the target data is compared to the population data.

$$\text{Score}_{\text{MIA}}(x; \theta) = \Pr_{z \sim \pi} (\text{LR}_\theta(x, z) \geq \gamma) \quad (4)$$

$$\text{LR}_\theta(x, z) = \frac{\mathcal{L}(\theta|x)}{\mathcal{L}(\theta|z)} \quad (5)$$

The adopted offline mode Algorithm 1 shrinks from the need to retrain reference models per query, thus relying on pretrained LLMs, which are computationally expensive to train. For most experiments, we used just one reference model ($k = 1$), thus demonstrating the power of RMIA attack and highlighting data leakage, especially from pretrained data.

---

**Algorithm 1** MIA score calculation with offline RMIA [10] adapted to LLMs.

---

**Input:** $k$ reference models $\Theta$, target sample $x$, threshold $\gamma$, scaling factor $\alpha$, population dataset $\pi$,
**Output:** $\text{Score}_{\text{MIA}}(x; \theta)$

1: Randomly choose a subset $Z$ from the population dataset
2: $C \leftarrow 0$
3: $\mathcal{L}(x)_{\text{OUT}} \leftarrow \frac{1}{k} \sum_{\theta' \in \Theta} \mathcal{L}(x|\theta')$
4: $\mathcal{L}(x) \leftarrow \frac{1}{2}((1 + \alpha)\mathcal{L}(x)_{\text{OUT}} + (1 - \alpha))$
5: $\text{Ratio}_x \leftarrow \frac{\mathcal{L}(x|\theta)}{\mathcal{L}(x)}$
6: **for** each sample $z$ in $Z$ **do**
7: $\quad \mathcal{L}(z) \leftarrow \frac{1}{k} \sum_{\theta' \in \Theta} \mathcal{L}(z|\theta')$
8: $\quad \text{Ratio}_z \leftarrow \frac{\mathcal{L}(z|\theta)}{\mathcal{L}(z)}$
9: $\quad$ **if** $\text{Ratio}_x/\text{Ratio}_z < \gamma$ **then**
10: $\quad\quad C \leftarrow C + 1$
11: $\quad$ **end if**
12: **end for**
13: **return** $\text{Score}_{\text{MIA}}(x; \theta) \leftarrow \frac{C}{|Z|}$

---

## 2. Privacy leakage

In Figure 5, we explore privacy leakage at different levels of DP guarantees $\varepsilon$. We find that even with small $\varepsilon$, there can still be a significant privacy leakage, especially when adaptations are trained with IID data.

## 3. RMIA Hyperparameters

We focus on the importance of $\gamma$, as $\alpha$ has a much more limited effect, and we set it to 0. Figure 6 shows the importance and $\gamma$ and suggests that $\gamma = 1$ is often the best choice. We omit it for simplicity, but a similar trend can be observed for the other settings.

## 4. Validation loss

Table 3 shows the validation loss at the end of the training for each adaptation on the selected hyperparameters.

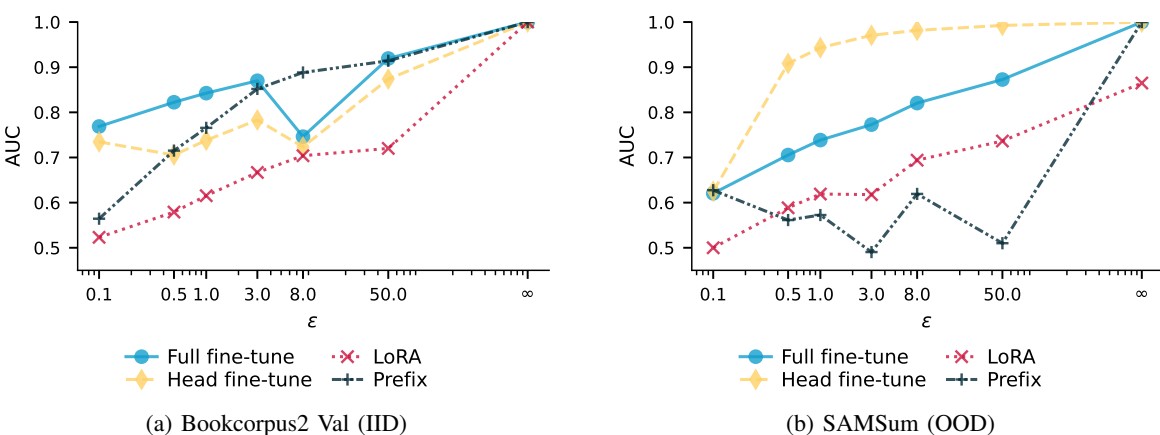

(a) Bookcorpus2 Val (IID)                    (b) SAMSum (OOD)

Figure 5: **The protection against MIA even for out-of-distribution (OOD) data requires tight privacy with $\varepsilon < 0.1$ for all the adaptations.** The x-axis represents the privacy budget with a log scale and the y-axis is the AUC score. The evaluation was done for $\varepsilon = \{0.1, 0.5, 1, 3, 8, 50\}$.

TABLE 3: **Validation loss values for the Pythia 1B model on different adaptation datasets.**

| Adaptation \ Dataset | samsum | | | german wiki | | | Bookcorpus2 Val | | | Bookcorpus2 Train | | | Github Val | | | Enron Val | | |
|---|---|---|---|---|---|---|---|---|---|---|---|---|---|---|---|---|---|---|
| | $\varepsilon = \infty$ | $\varepsilon = 8$ | $\varepsilon = 0.1$ | $\varepsilon = \infty$ | $\varepsilon = 8$ | $\varepsilon = 0.1$ | $\varepsilon = \infty$ | $\varepsilon = 8$ | $\varepsilon = 0.1$ | $\varepsilon = \infty$ | $\varepsilon = 8$ | $\varepsilon = 0.1$ | $\varepsilon = \infty$ | $\varepsilon = 8$ | $\varepsilon = 0.1$ | $\varepsilon = \infty$ | $\varepsilon = 8$ | $\varepsilon = 0.1$ |
| Prefix Tuning | 2.311 | 2.451 | 2.778 | 2.573 | 2.738 | 2.838 | 2.968 | 2.993 | 3.387 | 2.997 | 2.994 | 3.390 | 1.599 | 1.557 | 2.054 | 2.412 | 2.426 | 3.002 |
| LoRA | 2.313 | 2.462 | 2.761 | 2.578 | 2.737 | 2.801 | 2.951 | 3.007 | 3.013 | 2.979 | 3.002 | 3.003 | 1.558 | 1.572 | 1.558 | 2.394 | 2.402 | 2.403 |
| Full fine-tune | 2.251 | 2.457 | 2.759 | 2.511 | 2.726 | 2.747 | 2.934 | 2.999 | 3.028 | 2.960 | 2.995 | 3.020 | 1.598 | 1.566 | 1.577 | 2.375 | 2.397 | 2.413 |
| Head fine-tune | 2.354 | 2.454 | 2.761 | 2.574 | 2.731 | 2.756 | 2.949 | 3.007 | 3.339 | 2.966 | 3.002 | 3.332 | 1.577 | 1.573 | 1.750 | 2.409 | 2.403 | 2.536 |
| Average | 2.307 | 2.456 | 2.764 | 2.559 | 2.733 | 2.785 | 2.950 | 3.002 | 3.192 | 2.976 | 2.998 | 3.186 | 1.583 | 1.567 | 1.734 | 2.397 | 2.407 | 2.589 |

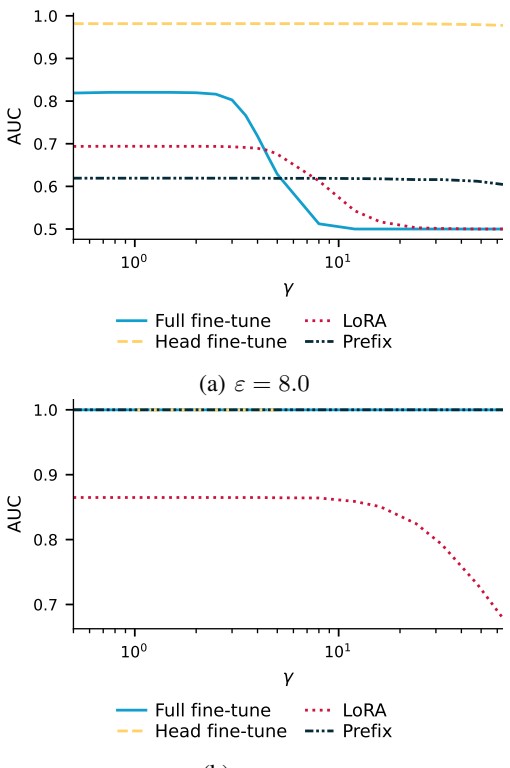

(a) $\varepsilon = 8.0$

(b) $\varepsilon = \infty$

Figure 6: $\gamma = 1$ **is a strong baseline.** We present the AUC using RMIA with different types values of $\gamma$ after adapting Pythia 1B on SAMSum. The evaluation was done for $\varepsilon = \{8, \infty\}$.