# OpenReview forum: "Auditing Empirical Privacy Protection of Private LLM Adaptations"
_NeurIPS.cc/2024/Workshop/SafeGenAi — SafeGenAi Poster_

### Official Review · Reviewer_qEUK · 2024-10-09
**Review of Submission90**

**Rating:** 7
**Confidence:** 4

**Review:**

Quality

Pros:

1. The paper presents a thorough empirical investigation into privacy risks associated with the adaptation of large language models, using state-of-the-art membership inference attacks, demonstrating that even a DP-trained model may have privacy risks.

2. The experimental setup is well-structured, evaluating different adaptation techniques (e.g., LoRA, prefix tuning, full fine-tuning) and privacy budgets.

3. The diverse range of datasets (in-distribution, out-of-distribution, and overlap) enhances the replicability and transparency of the study.


Cons:

1. The experiments are only conducted in one single model Pythia, which might restrict the conclusion

2. It is better if the authors provide more MIA methods to see whether only using SOTA MIA method can threaten the privacy model.

Clarity

Pros:
1. The paper clearly outlines the main research question.

2. Overall, the paper is well-written and easy to follow.


Cons:

1. The experimental settings might be too much. Maybe it is better to introduce one setting with one result for different experiments, which is easier to understand.

Originality

Pros:
1. The paper introduces a novel empirical evaluation of privacy risks associated with LLM adaptations, filling an important gap in understanding the practical implications of private adaptations.

2. The exploration of privacy leakage across different distributions (in-distribution, out-of-distribution, and overlapping data) provides a new perspective on the risks associated with LLM training and adaptation.


Cons:

1. The idea that DP-trained models might also have privacy risks has already been demonstrated.

Significance

Pros:
1. The findings are significant for the field of privacy-preserving machine learning, verifying the overlap between pre-trained data and fine-tuned data will influence the privacy adaptation.

---

### Official Review · Reviewer_NRH1 · 2024-10-09
**Reviewer NRH1**

**Rating:** 6
**Confidence:** 4

**Review:**

### Paper Summary
This paper investigates the issue of privacy leakage during the personalization fine-tuning process of large language models (LLMs). The study evaluates the effectiveness of membership inference attacks under different data adaptation strategies, particularly parameter-efficient fine-tuning methods (PEFT) such as Low-Rank Adaptation (LoRA) and prefix tuning. It finds that the risk of privacy leakage is lower when the adaptation data is more similar to the pre-training data. Additionally, the paper discusses the impact of different datasets and model architectures on privacy leakage.

### Strengths
1. **Originality and Practical Significance**: The paper addresses a research gap in LLM privacy protection, proposing innovative evaluation methods and privacy protection strategies with high originality and practical application value.
2. **Detailed Experimental Design**: The article systematically assesses privacy risks through a combination of multiple datasets and fine-tuning strategies, with rigorous experimental design and data supporting the conclusions.
3. **In-depth Analysis**: The paper provides a thorough analysis of the experimental results, discussing how different factors such as data distribution differences and fine-tuning techniques specifically affect privacy leakage.

### Weaknesses
1. **Insufficient Theoretical Support**: Although the experimental results are rich, the theoretical analysis is relatively weak, lacking a deep theoretical explanation of the mechanisms of privacy leakage.
2. **Single Privacy Attack Model**: The study mainly focuses on membership inference attacks and does not consider other types of privacy attacks, such as model inversion or attribute inference attacks.
3. **Generalizability Issues**: The research is based on specific datasets and models, which may limit the generalizability of the conclusions.

### Recommendations for Improvement
1. **Extend Theoretical Analysis**: It is recommended to add support from information theory or computational complexity theory to better explain and predict the impact of different adaptation strategies and data distributions on privacy leakage.
2. **Diversify Privacy Attack Assessments**: Introduce a wider variety of privacy attack models, such as model inversion attacks and attribute inference attacks, to comprehensively evaluate the privacy protection effectiveness of the model under different attacks.
3. **Enhance Generalizability Studies**: Conduct experiments across datasets in multiple languages and domains, and use LLMs with various architectures, to assess the generalizability and practicality of the proposed methods.
4. **Real-World Application Case Studies**: Consider adding case studies of practical application scenarios, particularly in sensitive areas such as healthcare or finance, to demonstrate the effectiveness and challenges faced by the methods in practical operations.